# Nano Groove and Prism-Structured Triboelectric Nanogenerators

**DOI:** 10.3390/mi14091707

**Published:** 2023-08-31

**Authors:** Resul Saritas, Majed Al-Ghamdi, Taylan Memik Das, Omar Rasheed, Samed Kocer, Ahmet Gulsaran, Asif Abdullah Khan, Md Masud Rana, Mahmoud Khater, Muhammed Kayaharman, Dayan Ban, Mustafa Yavuz, Eihab Abdel-Rahman

**Affiliations:** 1Department of Systems Design Engineering, University of Waterloo, Waterloo, ON N2L 3G1, Canada; skocer@uwaterloo.ca (S.K.); eihab@uwaterloo.ca (E.A.-R.); 2National Security Program King Abdulaziz City for Science and Technology, KACST, Riyadh 12354, Saudi Arabia; 3Department of Mechanical Engineering, University of Kirikkale, Kirikkale 71450, Turkey; taylandas@gmail.com; 4Department of Mechanical and Mechatronics Engineering, University of Waterloo, Waterloo, ON N2L 3G1, Canada; agulsaran@uwaterloo.ca (A.G.); myavuz@uwaterloo.ca (M.Y.); 5Department of Mechanical Engineering, University of Toronto, Toronto, ON M5S 1A1, Canada; rasho1959@gmail.com; 6Department of Electrical and Computer Engineering, University of Waterloo, Waterloo, ON N2L 3G1, Canada; aakhan@uwaterloo.ca (A.A.K.); mm4rana@uwaterloo.ca (M.M.R.); dban@uwaterloo.ca (D.B.); 7Department of Mechanical Engineering, King Fahd University of Petroleum and Minerals, Dhahran 31261, Saudi Arabia; mkhater@uwaterloo.ca

**Keywords:** triboelectric, nano features, nano groove and prism, PDMS

## Abstract

Enhancing the output power of triboelectric nanogenerators (TENGs) requires the creation of micro or nano-features on polymeric triboelectric surfaces to increase the TENGs’ effective contact area and, therefore, output power. We deploy a novel bench-top fabrication method called dynamic Scanning Probe Lithography (d-SPL) to fabricate massive arrays of uniform 1 cm long and 2.5 µm wide nano-features comprising a 600 nm deep groove (NG) and a 600 nm high triangular prism (NTP). The method creates both features simultaneously in the polymeric surface, thereby doubling the structured surface area. Six thousand pairs of NGs and NTPs were patterned on a 6×5 cm${}^{2}$ PMMA substrate. It was then used as a mold to structure the surface of a 200 µm thick Polydimethylsiloxane (PDMS) layer. We show that the output power of the nano-structured TENG is significantly more than that of a TENG using flat PDMS films, at 12.2 mW compared to 2.2 mW, under the same operating conditions (a base acceleration amplitude of 0.8 g).

## 1. Introduction

Demand for distributed electrical power has seen a significant growth as Internet of Things applications come to market creating demand for dense sensor networks made of many millions of nodes [1,2]. Localized power supply is an attractive approach to power those networks, which is driving a search for new power supply modalities. Waste mechanical vibrations is a promising distributed power source readily available in many environments [3,4].

Waste vibrations have been converted into electrical energy using piezoelectric [5], electromagnetic [6], and electrostatic nanogenerators [7]. All of these methods are potentially clean, highly efficient, and environmentally friendly [8]. A new type of electrostatic nanogenerators, triboelectric nanogenerators (TENGs), was introduced in 2012 [9]. TENGs convert mechanical vibrations into electrical energy by exploiting the triboelectric effect to provide the voltage source necessary for electrostatic induction [10,11,12]. Upon separation of a metallic and a dielectric film, the metal loses electrons, becoming positively charged while the dielectric gains electrons, becoming negatively charged [13,14]. The two charged films serve as a voltage source within a variable capacitor [15]. In turn, the capacitor drives a current between its plates as they move with respect to each other under the influence of mechanical vibrations [16].

TENGs can provide a power density in the range of mW/cm${}^{2}$ sufficient to power many autonomous sensor nodes and other self-powered portable electronic devices [17,18]. A variety of materials with opposite polarity have been implemented in TENGs to improve their efficiency, particularly polytetrafluoroethylene (PTFE) and poly-dimethylsiloxane (PDMS), which are frequently chosen as negative triboelectric material due to their high tendency to receive electrons [19,20,21].

The output power of TENGs has also been enhanced by creating nano structured triboelectric surfaces, thereby increasing the contact area between the two triboelectric films. This allows more freely moving electrons to leave the metallic film to the dielectric film resulting in a higher voltage drop between the two films and increasing the output power of the TENG [22]. Two classes of fabrication methods have been adopted to achieve this goal. The first creates those structures using oxides, by the etching of TiO_2_ nanowires and nano-sheets into Ti films [23], chemical synthesis of SiO_2_ nanoparticles on SiO_2_ films [24], and growing of ZnO nanorods on indium tin oxide (ITO) films [25], or metals, by etching of gold flowers into gold films [26]. The second creates nano-features, such as PDMS nanopillars [27], PDMS triangular pyramids, and PDMS triangular ridges [26,28,29], by molding polymers using standard imprint lithography (SIL).

Currently, the creation of those nano-features requires the use of cleanroom facilities, for electrochemcial deposition [30] or standard soft lithography [31], thereby imposing cost and access barriers on TENG technology. Alternatively available low-cost and rapid fabrication methods such as the CO${}_{2}$ laser ablation method are not able to produce features less than 50 µm, which limits the maximum surface area in turn power generation. In this study, we employ a novel and low-cost bench-top method to fabricate micro- and nano-features. The method allows us to introduce massive arrays of one centimeter-long nano triangular prisms (NTPs) and nano grooves (NGs) on a Poly(methyl methacrylate) (PMMA) substrate. It employs a micro-resolution 3D stage equipped with a micro-sized tip via a spring-damper system. In total, twelve thousand NGs and NTPs are patterned in a 6×5 cm${}^{2}$ PMMA substrate, which is used as a mold to transfer those features into PDMS substrates using spin coating. The fabricated PDMS films serve as the negative triboelectric layer while aluminum (Al) foil is used as the positive triboelectric layer. The output power of the fabricated TENG is compared to that of an identical TENG made using a flat surface PDMS layer.

## 2. The TENG

### 2.1. Fabrication of Nano-Structured Films

The fabrication method employs a micro-resolution needle mounted to a micro-resolution 3D stage via a spring-damper mechanism. As the needle traverses the polymeric substrate surface, non-uniformity in the morphology of the substrate surface generates time-variation in the magnitude and direction of the contact force, which in turn leads to needle vibrations. The spring-damper mechanism acts to dissipate those vibrations and stabilizes the contact force, thereby generating uniform features in the substrate. An optical camera equipped with a 6× objective lens and 2× optical magnifier is attached to the stage to monitor the patterning process. Figure 1a shows the fabrication setup used to pattern the PMMA substrate.

The desired pattern and a constant contact force are defined in an interface software SonoGuide, Version 2.42 [32] which translates it into a 3D trajectory written in machine language. The controller of the micro-stage follows that trajectory as it traverses the needle tip over the substrate surface while tracking the process using the optical camera. Upon first contact, the of Z-axis actuator commands a displacement z(t) resulting in the tip penetrating the substrate surface with an average displacement of x(t) and a spring compression force of
(1)F(t)=k(z(t)−x(t))
where *k* is the spring stiffness given as 100 N/m, acting on the tip.

The desired pattern is made of a massive array of uniform 1 cm long, 2.5 µm wide 600 nm deep NGs and 600 nm high NTPs were patterned. The feature dimensions, width and height, are controlled by varying the substrate-tip contact force while the length is commanded by the micro-stage to the desired length. The sharpness of the needle tip, and the hardness of the substrate determine whether patterning results in plastic deformations or chip removal is required. In this case, the contact force and the needle tip orientation with respect to the trajectory were calibrated to plastically deform the PMMA substrate. As the deformation created NGs in the surface, the material flowing out of them created NTPs on their edges, resulting in the simultaneous creation of two nano features in a single patterning event. The dual nano features increase the effective surface area, thus enhancing triboelectricity. The patterned PMMA substrate, shown in Figure 1b exhibiting an array of 6000 cm long NGs and NTPs, is then used as a mold to cast PDMS films.

The PDMS base and its curing agent are mixed at a weight ratio of 5:1 and spin-coated on the PMMA mold, Figure 1c. The mold and cast PDMS, Figure 1d, are placed for 5 min on a hot-plate for curing at 80 ${}^{\circ }$C. Multiple spin-coating at 500 RPM and curing cycles are undertaken to reach a thickness of 200 µm. The PDMS film is then peeled off from the mold resulting in the surface pattern shown schematically in Figure 1e. The surface topography of the fabricated film was measured using an atomic force microscope (Bruker AFM, Camarillo, CA, USA). Figure 1f shows the cross-sectional profile of the NGs and NTPs on the fabricated PDMS film. The bases of the NGs and NTPs were measured to be 2.5 µm wide and their apexes were measured to be 600 nm high.

### 2.2. Fabrication of the TENG

Figure 2a shows optical microscope (Nikon Inverted Microscope, Tokyo, Japan) images of uniformly patterned 2.5 µm wide and 600 nm deep and 1 cm long pairs of NGs and NTPs.The area shown, 4 × 3 mm${}^{2}$, exhibits 85 pairs of NGs and NTPs. A close-up view of the patterns in a 0.75×0.565 mm${}^{2}$ area is shown in Figure 2b and an AFM image of the plastically deformed NGs and flown NTPs in the area of 15 × 15 µm${}^{2}$ is shown Figure 2c.

The TENG consists of structural support elements and energy triboelectric harvesting elements. The former include two backbone PMMA plates, 8×8 cm${}^{2}$ in area and 0.87 cm in thickness, to provide structural support and an inertial energy harvesting mass. Four linear guides, 60 mm long and 3.2 mm in diameter steel rods, are placed at the four corners of the plates to maintain alignment between the plates as they oscillate against each other under host vibrations. Four concentric steel springs, 24 mm long with an outer diameter of 4.8 mm, inner diameter of 4.5 mm, and spring stiffness of 90 N/m, are housed around the linear guides to maintain separation between the plates. A 6×5 cm${}^{2}$ PMMA stage is attached to the bottom backbone plate using double-sided tape to control the capacitive gap. The harvesting elements are the PDMS film and two Al foil films, all with identical planar dimensions to the stage.

Using double-sided tape, an Al film is attached to the top backbone plate and another to the stage to serve as the positive triboelectric film and negative electrode, respectively. The flat bottom surface of the PDMS film is pressed onto the negative electrode while avoiding any air gaps at the interface.

Holes with a diameter of 3.2 mm were laser cut into the corners of the backbone plates within their inactive margins. The linear guides were inserted into the bottom backbone plate holes and fixed to it using super glue. Next, the springs were placed around the linear guides. Finally, the guides are inserted through the top backbone plate holes. The top plate was allowed to move freely up and down along the linear guides.

### 2.3. Working Principle

Figure 3 shows a schematic of the harvesting elements and their interaction throughout the harvesting cycle. The TENGs generate an alternating current as the external (host) vibrations cycle the harvesting elements through contact and separation states and the TENG undergoes the corresponding triboelectrification and electrostatic induction stages. Initially, the two electrodes are neutral and separate from each other. Under external vibrations, the top Al electrode and PDMS film come into contact. At this point, triboelectric charge generation begins as electrons are injected from the Al to the PDMS. As separation begins, the top plate moves upward under the influence of the springs’ restoring force acting against the internal electric field between the charges distributed on the two triboelectric surfaces. As a result, electrons flow from the bottom (negative) electrode to the top (positive) electrode under the influence of electrostatic induction between the PDMS film on the bottom electrode. This process terminates when the top plate reaches maximum separation distance. As it reverses direction and approaches the bottom plate, a current flows in the reverse direction as electrons flow from the top electrode under the influence of the electrostatic induction of the PDMS film on it to the bottom electrode. The cycle terminates with the triboelectric films coming back into contact.

## 3. Results and Discussion

Figure 4 shows the experimental setup of the triboelectric energy harvester. An electromagnetic shaker is driven by an amplifier (LABWORKS pa-138, Labworks Inc., Costo Mesa, CA, USA) and close loop controlled by a vibration controller (VR9500 Revolution, Michigan, CA, USA) using acceleration feedback. The output voltage and current of the TENGs are measured using an Oscilloscope (Tektronix-TDS2004C, Beaverton, OR, USA) via a high-input impedance (70 MΩ) probe. A low-noise current amplifier (SRS Stanford Research Systems-SR570 Preamplifier, Sunnyvale, CA, USA) is employed to measure the short-circuit current.

The flat and nano-structured PMDS films shown in Figure 5 serve as the negative triboelectric material. After peeling off the nano-structured PDMS film from the PMMA mold, its surface features are the reverse of those in the mold, namely six thousand NGs and NTPs each 1 cm long, 2.5 µm wide, and 600 nm deep and high, respectively.

The shaker was used to deliver base accelerations of the form:(2)a(t)=Acos(2πft)
with an amplitude of A=0.8 g and a frequency of f=8 Hz to the TENGs. The controller was used to maintain a constant acceleration amplitude as the resistive load was varied. To overcome the high impedance of the open-circuit TENG, the output voltage was measured through a 70 MΩ input impedance probe. Figure 5c compares the open-circuit voltage of the flat PMDS TENG to the nano-structured PMDS. It can be seen that the nano-structured film allowed the TENG to increase the peak-to-peak voltage from the range of $V_{pp}$ = 300–400 V to the range $V_{pp}$ = 1100–1400 V.

Figure 5d compares the short-circuit current through a 50 Ω resistor of the flat PMDS TENG to that of the nano-structured PMDS TENG. The peak-to-peak current generated by the nano-structured TENG was measured at $I_{pp}=200$ µA compared to $I_{pp}=40$ µA for the flat TENG. These results show that the nano-structured film increases the output voltage and current of the TENG four-fold compared to the flat film.

Nano-structuring the PDMS film increases its contact area with the Al films, thereby improving its ability to create a voltage difference across the TENG plates and, therefore, the output power. Specifically, the PDMS NTPs bend during contact with the Al film introducing sliding and friction between the two surfaces which allows thee PDMS to break the bonds between electrons on the Al surface more efficiently. Cycle-on-cycle variability in the impact, sliding, and friction processes result in the variability of open-circuit voltage observed in Figure 5c.

We compared the output power of the flat PDMS TENG to that of the nano-structured PDMS TENG by measuring the current and voltage they generated across a variable resistive load *R* connected between their terminals as per the schematic in Figure 6a. Parallel resistance imposed by the probe in evaluating the load resistance was accounted for the equivalent load. The current dropped and the voltage increased as the load resistance was increased from R=2Ω to R=120 MΩ. For the nano-structured TENG, the measured peak-to-peak current, the red line in Figure 6c, dropped from $I_{pp}=$ 245 µA to 8.8 µA, and the voltage, the blue line in Figure 6c, increased from $V_{pp}=0.5$ V to 1328 V.

The measured voltage and current were used to evaluate the instantaneous output power across the resisitive load as:(3)P(t)=V(t)I(t)

Figure 6b compares the maximum instantaneous power $P_{\max}$ of the flat TENG (red line) to that of the nano-structured TNEG (cyan line). We found that the optimal resistance of the flat TENG was $R_{opt}\approx 1.5$ MΩ while that of the nano-structured TENG was $R_{opt}\approx 2.0$ MΩ. We also found that the nano-structured TENG consistently outperformed the flat TENG irrespective of the load reaching an optimal maximum instantaneous output power of $P_{\max}=12.2$ mW compared to $P_{\max}=2.2$ mW for the flat TENG. This shows that introducing nano-structures into the surface of the PDMS film increased the TENG’s power generation capacity by more than five-fold.

We also examined the ability of the TENG to store energy by connecting it to a storage capacitor using the circuit layout shown in Figure 7a. A bridge rectifier, consisting of four Schottky diodes, was used to convert the alternating (AC) output current of the TENG to direct current (DC) to charge the capacitor. We measured the voltage across the capacitor as the harvester charged it by harvesting kinetic energy from the vibrations waveform described above.

Figure 7b shows the evolution over time of the voltage V(t) across five storage capacitors ranging in capacitance from C=0.1 µF to 4.7 µF. We found that the energy stored in the capacitor, evaluated as
(4)$$E=\frac{1}{2}CV^{2}$$
ranged from E=5.2 µJ in 4s for the C=0.1µF capacitor to E=18 µJ in 16 s for the C=3.3 µF capacitor. While neither the on-resistance losses of the diodes or the capacitor size were optimized, the results show that stored energy was delivered at a rate better than 1 µW.

Finally, a schematic of the TENG demonstration with LEDs is given in Figure 8a, and we showed the comparative functionality of the TENGs by deploying them to directly power arrays of series-connected red, white, yellow, and blue LEDs, illustrated in Figure 8b. The array of LEDs was inserted into the energy harvesting circuit described above in place of the storage capacitor. The nano-structured PDMS TENG was able to light up a bank of 180 red, white, yellow, and blue LEDs shown in Figure 8c, whereas the maximum array size the flat PDMS TENG was able to light up was 67 LEDs.

## 4. Conclusions

We introduced a novel, rapid, uniform, and low-cost process for high-speed (1 mm/s) fabrication of long (1 cm) nano-structures (NGs and NTP) on the surface of PMMA substrates without a need for cleanroom facilities. We used this process to fabricate PMMA molds which, in turn, were deployed to produce nano-structured thin PDMS films. Those films were then employed as the negative triboelectric material in TENGs. Surface nano-structuring was found to increase the output power from 2.2 mW for flat PDMS TENGs to 12.2 mW for nano-structured PDMS TENGs. Ultimately, nano-structured TENGs proved to be a more effective power source able to light an array of 180 LEDs when the flat TENGs were only able to light 67 LEDs.

## Figures and Tables

**Figure 1 micromachines-14-01707-f001:**
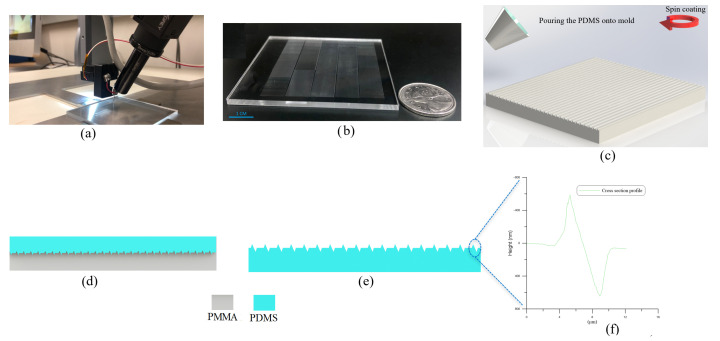
(**a**) The fabrication setup. (**b**) An array of 6000 NGs and NTPs on a 6×5 cm${}^{2}$ area of a PMMA substrate. (**c**) Spin coating of PDMS onto the PMMA mold. (**d**) Curing of PDMS. (**e**) Peel off of the PDMS film from the mold. (**f**) A cross-sectional profile of the film obtained using AFM microscopy. The profile shows the NGs and NTPs.

**Figure 2 micromachines-14-01707-f002:**
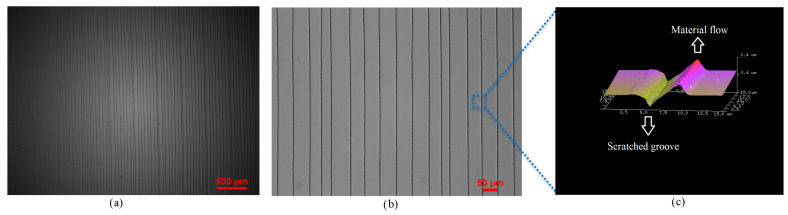
(**a**) A microscopic image of an array made of 85 pairs of NGs and NTPs in a 4×3 mm${}^{2}$ area. (**b**) A close-up microscopic image of a 750×565 µm${}^{2}$ area. (**c**) AFM image of the NGs and NTPs in a 0.75×0.565 mm${}^{2}$ area illustrating plastic deformation and material flow.

**Figure 3 micromachines-14-01707-f003:**
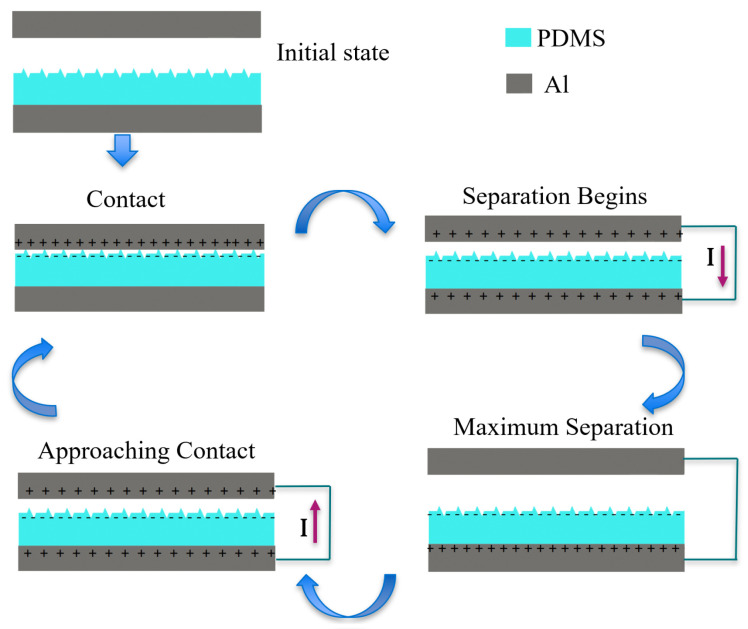
Triboelectric cyclic harvesting of mechanical energy.

**Figure 4 micromachines-14-01707-f004:**
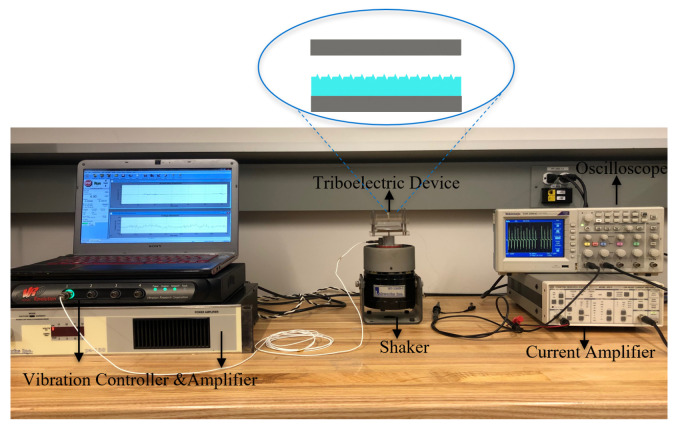
The experimental setup mounts the TENG on electromagnetic shaker and uses a current amplifier and an oscilloscope to measure the output signal.

**Figure 5 micromachines-14-01707-f005:**
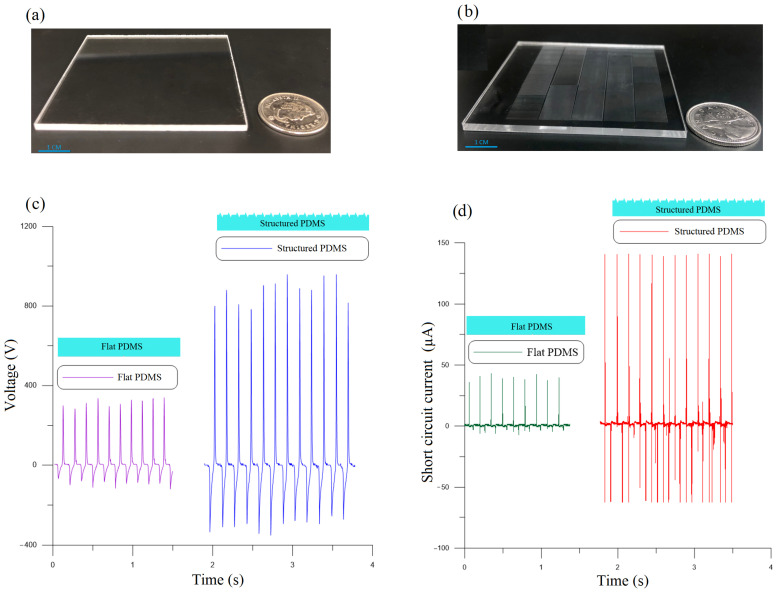
(**a**) The flat and (**b**) nano-structured molds used to cast the PDMS films. (**c**) Open-circuit voltage and (**d**) short-circuit current of the flat and nano-structured TENGs.

**Figure 6 micromachines-14-01707-f006:**
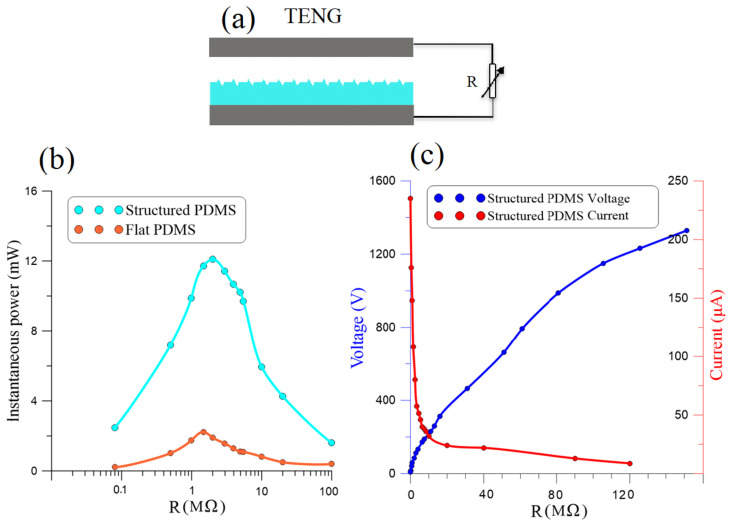
(**a**) A schematic illustrating measurement of the voltage and current generated by the TENG across a resistive load *R*. (**b**) Comparison of the output power of the nano-structured to the flat TENGs across *R*. (**c**) The measured voltage and current generated by the nano-structured TENG across *R*.

**Figure 7 micromachines-14-01707-f007:**
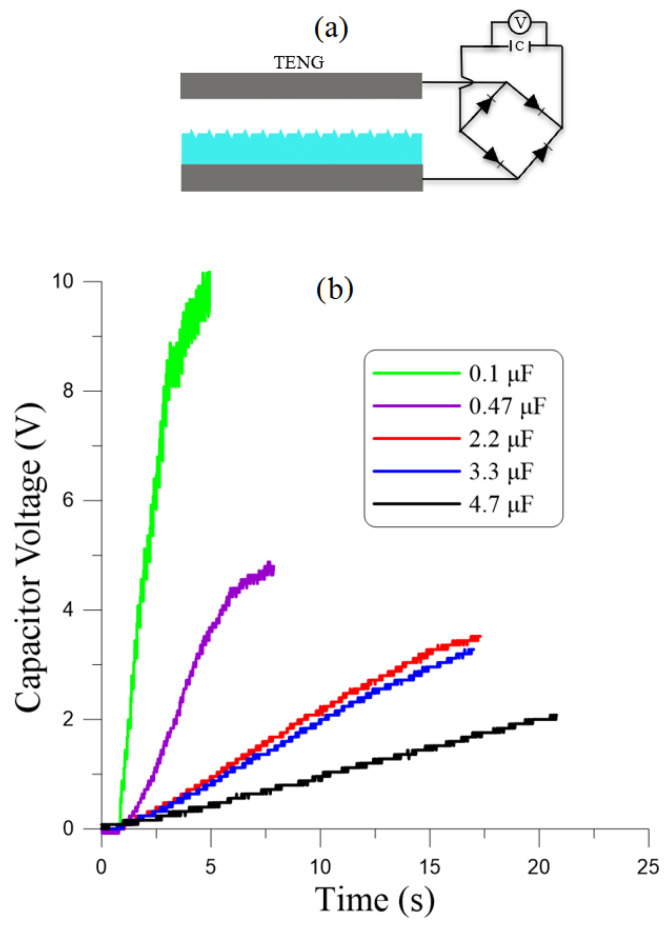
(**a**) A schematic of the TENG and its charging circuit comprising a capacitor and a bridge rectifier. (**b**) Measured capacitor voltage as a function of time for five capacitance values.

**Figure 8 micromachines-14-01707-f008:**
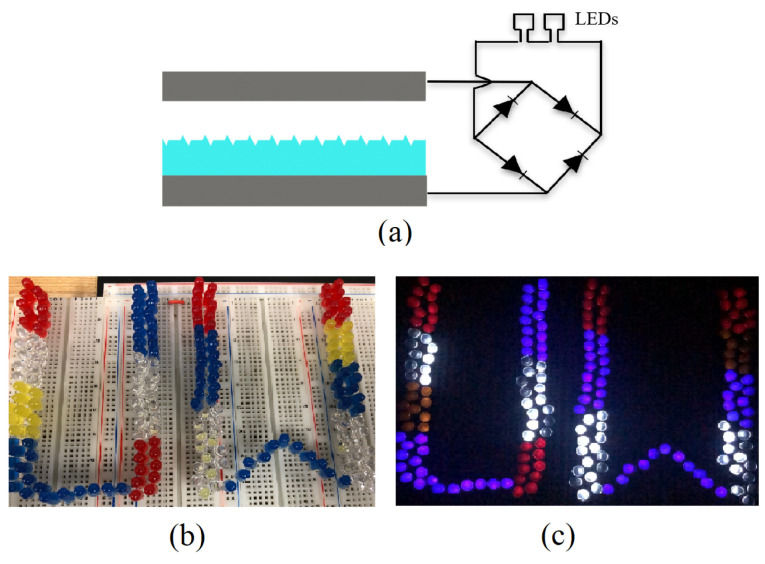
(**a**) A schematic of the TENG demonstration where they were deployed to power an array of series-connected red, white, yellow, and blue LEDs via a bridge rectifier. Images of an array of 180 LEDs while their power source, a nano-structured TENG, was (**b**) off and (**c**) on.

## Data Availability

Not applicable.

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
