# Peer review of "Nano Groove and Prism-Structured Triboelectric Nanogenerators"

_micromachines, 2023, doi:10.3390/mi14091707_

Round 1
Reviewer 1 Report
In this work, the authors provided an important work, which significantly improve the output performances of triboelectric nanogenerators via surface microarchitecture. Therefore, this work can be accepted after a minor revision. The detailed comments are as follows.
1. Since the resistance of the probe of the oscilloscope is only 70 MΩ, the measured power in Figure 6 is not accurate. Please revise the related content in the manuscript.
2. Please cite more similar references, such as https://www.nature.com/articles/s41467-022-33454-y and https://pubs.rsc.org/en/content/articlelanding/2019/ta/c9ta01956a, to improve the strength of the manuscript.
The quality of English language is good.
Author Response
On behalf of the authors, I would like to thank the reviewer for their efforts and constructive feedback. Your evaluation and suggestions have helped us improve the accuracy and clarity of our manuscript. We have made the following change in response to your recommendations:
- Since the resistance of the probe of the oscilloscope is only 70 MΩ, the measured power in Figure 6 is not accurate. Please revise the related content in the manuscript.
Thank you very much for your valuable feedback. Indeed, we accounted for parallel resistance imposed by the probe in evaluating the equivalent load resistance. We have added a clarification to this effect in the manuscript on lines 179 and 180.
- Please cite more similar references, such as https://www.nature.com/articles/s41467-022-33454-y and https://pubs.rsc.org/en/content/articlelanding/2019/ta/c9ta01956a, to improve the strength of the manuscript.
Thank you very much for this suggestion. We have now cited those papers in the introduction.

Reviewer 2 Report
The article presents a study focused on enhancing the output power of triboelectric nanogenerators (TENGs) through the creation of micro or nano-features on polymeric triboelectric surfaces. The authors introduce a new fabrication method named dynamic Scanning Probe Lithography (d-SPL), which is utilized to create arrays of consistent nano-features measuring 1 cm in length and 2.5 μm in width. The fabrication process is detailed, involving the creation of six thousand pairs of NGs and NTPs on a PMMA substrate, which is subsequently used as a mold to pattern the surface of a PDMS layer. The core finding of the study: the output power of the resulting nano-structured TENG is significantly greater than that of a TENG utilizing flat PDMS films. Overall, the article succinctly communicates the purpose, methodology, and key results of the research. It outlines the innovation of the dynamic Scanning Probe Lithography method for fabricating enhanced nano-features on polymeric surfaces and emphasizes the substantial improvement in TENG output power achieved through this approach.
There are two minor changes to be done:
Line 14: Interest of things
Line 93: Spinning rpms should be mentioned
Author Response
On behalf of the authors, I would like to thank the reviewer for their efforts and constructive feedback. Your evaluation and suggestions have helped us improve the accuracy and clarity of our manuscript. We have made the following change in response to your recommendations:
- Line 14: Interest of things.
Thank you very much. We meant “Internet of Things”. We have corrected this typo. Line 14 and 15.
- Line 93: Spinning rpms should be mentioned.
Thank you very much for this suggestion. We have added the spinning rpm to manuscript on line 95.
